# Advances in ^177^Lu-PSMA and ^225^Ac-PSMA Radionuclide Therapy for Metastatic Castration-Resistant Prostate Cancer

**DOI:** 10.3390/pharmaceutics14102166

**Published:** 2022-10-11

**Authors:** Sui Wai Ling, Erik de Blois, Eline Hooijman, Astrid van der Veldt, Tessa Brabander

**Affiliations:** 1Department of Radiology & Nuclear Medicine, Erasmus MC, 3015 GD Rotterdam, The Netherlands; 2Department of Hospital Pharmacy, Erasmus MC, 3015 GD Rotterdam, The Netherlands; 3Department of Medical Oncology, Erasmus MC Cancer Institute, 3015 GD Rotterdam, The Netherlands

**Keywords:** actinium-225, lutetium-177, prostate-specific membrane antigen (PSMA), metastatic castration-resistant prostate cancer (mCRPC), radionuclide therapy (RNT)

## Abstract

For patients with metastatic castration-resistant prostate cancer (mCRPC), the survival benefit of classic treatment options with chemotherapy and drugs targeting androgen signaling is limited. Therefore, beta and alpha radionuclide therapy (RNT) have emerged as novel treatment options for patients with mCRPC. Radioligands target the prostate-specific membrane antigen (PSMA) epitopes, which are upregulated up to a thousand times more in prostate cancer cells compared to the cells in normal tissues. For this reason, PSMA is an excellent target for both imaging and therapy. Over the past years, many studies have investigated the treatment effects of lutetium-177 labeled PSMA (^177^Lu-PSMA) and actinium-225 labeled PSMA (^225^Ac-PSMA) RNT in patients with mCRPC. While promising results have been achieved, this field is still in development. In this review, we have summarized and discussed the clinical data of ^177^Lu-PSMA and ^225^Ac-PSMA RNT in patients with mCRPC.

## 1. Introduction

Prostate cancer (PCa) is one of the most common types of cancer in men, with 1.4 million patients estimated in 2020 worldwide [1,2]. As the general population continues to grow with increasing life expectancy, the number of patients with PCa is expected to further increase [2]. After primary diagnosis, approximately 10–20% of patients with PCa will have castration-resistant prostate cancer (CRPC) within five years and more than 80% of patients with CRPC will have metastatic disease [3]. CRPC is defined as disease progression after an initial response to androgen deprivation therapy. Currently, available therapeutic options for mCRPC are systemic treatments such as chemotherapy (docetaxel or cabazitaxel), androgen receptors or androgen receptor signaling inhibitors (abiraterone or enzalutamide), radium-223 treatment (for patients with bone metastases only), immunotherapy (sipuleucel-T or ipilimumab) and poly ADP-ribose polymerase (PARP) inhibitors (olaparib and rucaparib) [4]. However, the survival benefit is limited to only a few months [5,6]. In the past few years, promising results have been reported in patients treated with beta and alpha radionuclide therapy (RNT) using the prostate-specific membrane antigen (PSMA) labeled with lutetium-177 (Lu-177) or actinium-225 (Ac-225), respectively [7,8,9,10,11,12,13,14,15,16,17,18,19,20,21,22,23,24,25,26,27]. These treatments provide patients with mCRPC new options to improve survival and quality of life.

Prostate-specific membrane antigen (PSMA) is a type II 750 amino acid transmembrane glycoprotein (also known as folate hydrolase I or glutamate carboxypeptidase II) with both an intracellular and extracellular domain [28,29,30,31]. PSMA is expressed on benign prostate epithelium and on prostate cancer cells. There is also expression of PSMA in other tissues, such as the kidneys, small intestine and the salivary glands [32,33]. However, the expression on prostate cancer cells is a thousand-fold higher than expression on normal tissues, and in particular, higher PSMA expression was associated with more aggressive prostate tumors, such as hormone-refractory cancers [34,35]. Therefore, PSMA is a promising target for both imaging and therapy in patients with prostate cancer.

The concept of theranostics was first mentioned by John Funhouser (CEO of PharmaNetics) in 1998 [36]. The term “theranostics” refers to the possibility for both diagnostic and therapeutic use depending on the radionuclide and the ligand.

The classic PSMA-11 is only suitable for diagnostic use. However, the other PSMA ligands, such as PSMA-617 and PSMA Imaging and Therapy (I&T), are suitable for therapeutic use and are the most-investigated PSMA ligands for therapy with comparable behavior in vivo, toxicity profile and effectivity in patients with mCRPC [37,38,39,40]. In clinical practice, PSMA is usually labeled with the radionuclides Lu-177 and Ac-225 for therapeutic purposes.

Lu-177 is a beta-emitter with a half-life of 6.7 days. Through decay, Lu-177 emits medium-energy beta particles with an energy of 490 KeV and a tissue range of 0.7 to 2.1 mm [41]. In addition, Lu-177 emits gamma rays on decay with most abundant peaks of 112.9 and 208.4 KeV [41]. These gamma rays enable image acquisition and dosimetry calculation using a gamma camera [42,43,44].

Ac-225 is an alpha-emitter with a half-life of 9.9 days and decays via a cascade of six short-lived radionuclide daughters to near-stable bismuth-209. When decay occurs, Ac-225 emits four alpha particles in total with energies ranging from 5.8 to 8.4 MeV and a tissue penetration of 47 to 85 µm [45]. The cascade also includes two beta disintegrations of 1.6 and 0.6 MeV. Furthermore, gamma coemissions are generated from disintegrations of francium-221 and bismuth-213, which may be used for imaging [45].

After intravenous administration, PSMA ligands labeled with Ac-225 or Lu-177 accumulate in prostate cancer cells, which leads to the induction of DNA damage and ultimately to cell death [46]. The use of alpha emission offers advantages over beta emission due to the high linear energy transfer (LET) and the limited range in tissue [47]. The high LET selectively kills tumor cells through DNA double-strand and DNA cluster breaks, while sparing healthy tissue [45,48]. Most of the metastases of PCa are commonly found in lymph nodes and bones with off-target radiation that affects intra- and retroperitoneal fat and cortical bone. Both fat and bone are relatively radioresistant and therefore of minimal clinical relevance despite the decreased off-target radiation of Ac-225 compared to Lu-177 [49].

This review aims to summarize the clinical data of ^177^Lu-PSMA and ^225^Ac-PSMA RNT in patients with mCRPC and to provide future perspectives on RNT in prostate cancer patients, such as applying these therapies in earlier stages or exploration of other potential radionuclides.

## 2. ^177^Lu-PSMA

### 2.1. Response after ^177^Lu-PSMA

In the past decade, several studies have been performed to investigate the potential treatment effects of ^177^Lu-PSMA. In 2015, the first proof of concept in humans was performed by Weineisen et al. [50]. Two patients with mCRPC and multiple metastases in bone and lymph nodes were treated with 5.7 and 8.0 GBq. The therapy was well-tolerated in both patients without significant hematological or renal toxicity. In the following years, several groups [7,8,9,10,11,12,13,14,15,16] have treated patients with mCRPC with ^177^Lu-PSMA (summarized in Table 1). In 2019, Heck et al. [9] reported their clinical experience with [^177^Lu]Lu-PSMA-I&T in a larger cohort (n = 100) of patients with mCRPC. In this study, a total of 319 cycles were given to 100 patients with a median of two cycles per patient. Overall, treatment was well tolerated and a prostate-specific antigen (PSA) response was reported in the majority of patients.

In the same year, von Eyben et al. [11] performed a study of ^177^Lu-PSMA in patients with predominant lymph node metastasis (LNM). A total of 45 patients were included: 10 patients had local lesions, 29 patients had regional LNM, 30 patients had nonregional LNM and 10 patients had LNM with one or two bone metastases. Patients were treated with a median of three cycles with a median cumulative activity of 14.5 GBq. After ^177^Lu-PSMA, the median PSA declined with a maximum of 92%. In patients with only LNM, a larger PSA decline was observed as compared to patients with LNM and one or two bone metastases (maximum percentile PSA 77% vs. 19%). In 2020, Gallyamov et al. [13] performed a study in 110 patients with mCRPC who received two to five cycles with ^177^Lu-PSMA. In this study, therapeutic outcomes were categorized into three groups: definitive response with PSA decline >50% (n = 60), definitive failure or response with PSA decline <50% (n = 38) and therapy discontinuation due to adverse event or unrelated medical or nonmedical issues (n = 5). In seven patients, therapeutic outcomes could not be appropriately assessed. A recent systematic review [51], which included 24 articles and 1192 patients, evaluated the efficacy of ^177^Lu-PSMA. The estimated proportion of patients who were treated with either ^1^[^177^Lu]Lu-PSMA-617 or [^177^Lu]Lu-PSMA-I&T who had a PSA decrease of ≥50% was 0.44, which corresponded with the aforementioned studies.

Recently, results of the VISION trial [16], the largest phase III trial on PSMA therapy, were published. In this randomized trial, patients with progressive disease of PSMA-positive mCRPC were treated with ^177^Lu-PSMA RNT and standard of care (SOC) or with SOC alone. The primary objective of this trial was to compare the radiological progression-free survival (PFS) and overall survival (OS) in the two treatment arms. Patients treated with ^177^Lu-PSMA received [^177^Lu]Lu-PSMA-617 every six weeks for at least four cycles (up to six cycles) with an activity of 7.4 GBq per cycle. A total of 831 patients were included in this study, and the patients were randomly assigned to either the ^177^Lu-PSMA group (n = 551) or the SOC group (n = 280). The median follow-up was 20.9 months. Analysis of the primary endpoints showed a median imaging-based PFS and OS of 8.7 and 15.3 months for the ^177^Lu-PSMA group compared to 3.4 and 11.3 months for the SOC group, respectively (hazard ratio (HR) for progression or death was 0.40 and HR for death was 0.62). Analysis of secondary endpoints showed a median time to the first symptomatic skeletal event or death of 11.5 months for the ^177^Lu-PSMA group and 6.8 months for the SOC group. Response assessment according to RECIST 1.1 showed CR in 17 (9%) out of 184 patients in the ^177^Lu-PSMA group and in none of the 64 patients in the SOC group. PR was observed in 77 (42%) out of 184 patients in the ^177^Lu-PSMA group and in 2 (3%) out of 64 patients in the SOC group. A PSA decline of ≥50% and ≥80% was observed more frequently in the ^177^Lu-PSMA group compared to the SOC group. Grade 3 or higher toxicity was more frequently observed in the ^177^Lu-PSMA group compared to the SOC group (52.7% vs. 38%, respectively).

As more data on the efficacy of ^177^Lu-PSMA become available, the next important step is to investigate the place of ^177^Lu-PSMA for standard of care. The TheraP trial, a phase II trial lead by Hofman et al. [15], assessed the efficacy and safety of cabazitaxel as compared to ^177^Lu-PSMA in patients with mCRPC and previous treated with docetaxel. Among the 89 patients who were treated with ^177^Lu-PSMA, 45 patients completed protocol therapy with a median of five cycles. Among the 85 patients who were treated with cabazitaxel, 31 patients completed all ten planned cycles with a median of eight cycles. PSA decline >50% from baseline was observed more frequently in patients who received ^177^Lu-PSMA compared to cabazitaxel (66% vs. 37%, respectively). Progression analysis was assessed in 173 patients (n = 90 for ^177^Lu-PSMA and n = 83 for cabazitaxel) and showed that ^177^Lu-PSMA resulted in an improved PFS, radiographic PFS and PSA PFS (HR 0.63, 0.64 and 0.60, respectively) compared to cabazitaxel. In addition, in 78 patients with measurable disease by RECIST criteria at baseline, the objective response rate was higher in the ^177^Lu-PSMA group compared to the cabazitaxel group (49% vs. 24%, respectively). Safety analyses were assessed in 183 patients. Grade 3 or 4 toxicity occurred in 32 (33%) out of 98 patients in the ^177^Lu-PSMA group compared to 45 (53%) out of 85 patients in the cabazitaxel group. Grade 3 or 4 thrombocytopenia was observed more frequently in the ^177^Lu-PSMA group (11% vs. 0%). On the contrary, grade 3 or 4 neutropenia was observed more frequently in the cabazitaxel group (13% vs. 4%). Febrile neutropenia was only observed in the cabazitaxel group (8% vs. 0%).

### 2.2. PFS and OS after ^177^Lu-PSMA

Median OS and median PFS of ^177^Lu-PSMA RNT have been assessed in several studies (Table 1). In the study of Heck et al. [9], an additional analysis showed that the presence of visceral metastasis, younger age and rising lactate dehydrogenase (LDH) was associated with shorter PFS when treated with ^177^Lu-PSMA RNT. Factors associated with a shorter OS were primary metastatic disease, presence of visceral metastasis, younger age, rising PSA, rising alkaline phosphatase (ALP) and rising LDH. An additional analysis of Von Eyben et al. [11] demonstrated that a cumulative ^177^Lu-PSMA activity ≥14.8 GBq was associated with a longer PFS (determined by PSMA PET/CT using the PET response criteria for solid tumors (PERCIST)), and patients with only LNM showed a better OS compared to patients with LNM and one or two bone metastases.

Barber et al. [10] assessed PFS and OS in patients who were previously treated with taxane-based chemotherapy compared to patients without previous chemotherapy. In the entire cohort, prior taxane-based chemotherapy was not predictive for OS or radiographic PFS.

Bulbul et al. [14] showed a significantly longer OS and PFS in patients with a PSA decrease of ≥50% who had a response to ^177^Lu-PSMA RNT (21.8 and 11.7 months, respectively) compared to patients with a PSA decrease of ≥50% who did not have a response to ^177^Lu-PSMA RNT after the first cycle (13.7 and 5.6 months, respectively). Patients with any PSA decrease had a significantly longer OS and PFS (19.1 and 10.6 months, respectively) compared to patients who did not have any PSA decrease after the first cycle (13.8 and 4.0 months, respectively).

Furthermore, the survival analysis of a recent meta-analysis [51] showed pooled hazard ratios (HRs) of 0.29 and 0.67 for OS in patients with any PSA decline and PSA decline >50%, respectively. The pooled HR of the PFS in patients with a PSA decline >50% was 0.53.

Rosar et al. [52] investigated the value of early molecular imaging response assessment based on total viable tumor burden and its association with the OS. Univariate analysis showed that both biochemical and molecular imaging response assessments and ALP and Eastern Cooperative Oncology Group (ECOG) were significantly associated with OS. Multivariate analysis showed that molecular imaging response assessment, high ALP levels ≥220 U/L and an ECOG ≥ 2 remained independent predictors of OS, with HRs of 2.76, 3.08 and 2.21, respectively.

### 2.3. Toxicity after ^177^Lu-PSMA

Overall, therapy with ^177^Lu-PSMA was well-tolerated. The most common toxicity was grade 1/2 xerostomia. Other observed toxicities are summarized in Table 2.

Only one study [13] focused on the renal function in patients with mCRPC who were >3 months post-therapy. On clinical review, none of the patients experienced acute kidney injury (AKI). Within four to six weeks after treatment, acute kidney disease (AKD) was observed in two patients with a significant decrease in estimated glomerular filtration rate (eGFR) of 31% and 24%. The eGFR of both patients recovered within one month. Chronic kidney disease (CKD) was observed in three patients with a significant eGFR decrease of 20%, 22% and 23%. All patients who developed an AKD or CKD had previously been treated with both ADT and taxane-based chemotherapy before ^177^Lu-PSMA. No significant association was found between cumulative Lu-177 activity and eGFR changes.

## 3. ^225^Ac-PSMA

### 3.1. Response after ^225^Ac-PSMA

The first studies that were performed to investigate the potential treatment effects of ^225^Ac-PSMA RNT were performed by Kratochwil et al. in a small number of patients using [^225^Ac]Ac-PSMA-617 [17,18,19]. In 2016, the first two patients were treated in an experimental setting. The first patient was treated with three cycles of 9–10 MBq (100 kBq/kg) of [^225^Ac]Ac-PSMA-617 bimonthly. Two months after the third administration, all previously PSMA-positive lesions were not visible anymore on the PSMA PET/CT, and PSA decreased from more than 3000 ng/mL to 0.26 ng/mL. After a consolidation cycle of 6 MBq, the PSA declined further, to less than 0.1 ng/mL. The second patient was treated with three cycles of 6.4 MBq (100 kBq/kg) bimonthly. After two cycles, there was partial response on PSMA-PET/CT, and after three cycles, complete remission. PSA declined to less than 0.1 ng/mL. These patients showed a remarkable benefit from this treatment [17]. In 2017, the same group performed a retrospective analysis regarding toxicity and treatment response of 14 patients who were treated with different amounts of activity. The included patients were all heavily pretreated and were treated with different dose levels (50 kBq/kg, 100 kBq/kg, 150 kBq/kg and 200 kBq/kg) of [^225^Ac]Ac-PSMA-617. In a retrospective analysis, a treatment activity of 100 kBq/kg of [^225^Ac]Ac-PSMA-617 per cycle was considered a reasonable trade-off between toxicity and biochemical response [18]. However, a prospective phase I study is still lacking. Eventually, the results of 40 patients treated with ^225^Ac-PSMA RNT were reported by Kratochwil et al. [19]. A dose level of 100 kBq/kg was given to these patients every two months. Five patients discontinued treatment because they either had no response or early PSA relapse and four patients discontinued treatment because they experienced intolerable xerostomia or dysgeusia. PSA response is summarized in Table 3 [19].

Treatment responses of ^225^Ac-PSMA RNT in patients with mCRPC have also been evaluated by several other clinical research groups [20,21,22,23,27]. In the majority of studies, patients were treated with [^225^Ac]Ac-PSMA-617. However, recently, a retrospective analysis has been performed by Zacherl et al. including 14 patients with advanced mCRPC who were treated with [^225^Ac]Ac-PSMA-I&T. This study is the first study to present clinical data of patients treated with [^225^Ac]Ac-PSMA-I&T. A total of 11 (79%) out of the 14 patients were previously treated with ^177^Lu-PSMA. A total of 34 cycles of [^225^Ac]Ac-PSMA-I&T were administered with a mean activity of 7.8 MBq. A total of 7 (50%) out of the 14 patients showed a PSA decline ≥50% and 11 (79%) of the 14 patients showed any PSA decline [22].

A recent systematic review [53] showed the pooled data of three studies [19,21,23], which are also included in this review. The pooled proportion of patients with any PSA decline and a PSA decline >50% was 83% and 59%, respectively. PSA progression was observed in 21% of the pooled population. Molecular response using Gallium-68 labeled-PSMA PET/CT scan was observed in 17% of the pooled population.

Another recent systematic review [54], which included nine studies, showed comparable treatment response with a pooled proportion of patients with any PSA decline and a PSA decline >50% of 84% and 61%, respectively.

Three studies have also assessed the impact of ^225^Ac-PSMA RNT on quality of life (QoL). The first study was performed by Feuerecker et al. [24], who investigated the efficacy and adverse events of ^225^Ac-PSMA in patients with late-stage mCRPC after failure of ^177^Lu-PSMA. No measurable changes of the global health status or QoL were observed after the first or second cycle of ^225^Ac-PSMA using the European Organization for Research and Treatment for Cancer Quality of Life Questionnaire (EORTC-QLQ30).

The second study evaluated the efficacy of ^225^Ac-PSMA and impact on QoL in patients with advanced mCRPC and explored predictive biomarkers on pretherapeutic metastatic tissue biopsies [25]. QoL outcomes showed a clinically relevant decrease in pain symptoms with a corresponding reduced use of analgesics. Moderate and large improvements in the scores of physical and role functioning scales were observed at the end of therapy and at 12 months follow-up, respectively. Overall, a moderate improvement in global health state was achieved.

Finally, Satapathy et al. [26] assessed the impact of ^225^Ac-PSMA on QoL of patients with mCRPC using the National Comprehensive Cancer Network Functional Assessment of Cancer Therapy-Prostate Symptom Index 17 (NCCN-FACT-FPSI-17) questionnaire. Eleven heavily pretreated mCRPC patients were included in this study. A total of 25 cycles of ^225^Ac-PSMA were administered with a median cumulative activity of 8.3 MBq. QoL improved significantly when comparing pre- and post-therapy NCCN-FACT-FPSI-17 scores (29.8 vs. 41.3, respectively). Significant improvement was observed in pain, difficulty in urination, bone pain, fatigue and restriction in physical activity.

### 3.2. PFS and OS after ^225^Ac-PSMA

Only a few groups have assessed the PFS and OS after ^225^Ac-PSMA RNT [18,21,23]. These results of PFS and OS are summarized in Table 3. A recent systematic review [53] performed a survival analysis of these three studies and showed a pooled median OS and PFS of 17 and 12 months, respectively. Survival probability of the pooled proportion was 81% with a median follow-up duration of ten months. Furthermore, multivariate analysis by Yadav et al. [23] showed that a PSA increase of >25% is associated with poorer OS as compared to no PSA increase (8.5 vs. 17 months, HR 12.2). PFS was significantly lower in patients who did not show any PSA response compared to patients with any PSA response. In another recent systematic review [54], pooled analyses showed comparable median OS and PFS (11.77 and 9.15 months, respectively).

### 3.3. Toxicity after ^225^Ac-PSMA

The most common toxicity in patients treated with ^225^Ac-PSMA was xerostomia. Other toxicities are summarized in Table 2. Overall, the frequency and severity of most toxicities were quite comparable between ^225^Ac-PSMA and ^177^Lu-PSMA. Notably, toxicities in the critical organs such as the salivary glands and kidneys seem to be more frequent and more severe after ^225^Ac-PSMA.

## 4. Combination Therapy

### ^225^Ac-PSMA in Combination with ^177^Lu-PSMA Therapy

Despite the promising results of ^177^Lu-PSMA and ^225^Ac-PSMA RNT, up to 30% of the patients who are treated with ^177^Lu-PSMA monotherapy did not have a tumor response or develop resistance after an initial response. Biochemical and molecular response were achieved in 62.8% and 74%, respectively, of the patients who were treated with ^225^Ac-PSMA monotherapy [55,56,57]. For these patients, tandem therapy with ^177^Lu-PSMA combined with low-activity ^225^Ac-PSMA might be an alternative treatment option. Khreish et al. [55] evaluated the treatment response and severity of xerostomia of tandem therapy. Patients received one single cycle of [^225^Ac]Ac-PSMA-617/[^177^Lu]Lu-PSMA-617 tandem therapy, starting with [^225^Ac]Ac-PSMA-617 with a median administered activity of 5.3 MBq and followed by [^177^Lu]Lu-PSMA-617 with a median administered activity of 6.9 GBq in the same week (usually in consecutive days). According to Prostate Cancer Clinical Trials Working Group 3 criteria (PCWG3), partial response (PR), stable disease (SD) and progressive disease (PD) were observed in 10, 8 and 2 out of 20 patients, respectively. In addition, any PSA decline was observed in 16 (80%) out of 20 patients and a PSA decline >50% was observed in 13 (65%) out of 20 patients. With a median follow-up duration of 22 weeks, the median PFS and OS were 19 and 38 weeks, respectively. Overall, tandem therapy was well-tolerated without any serious acute adverse events. Grade 1 and 2 xerostomia were observed in eight and five patients, respectively. No grade 3 or 4 xerostomia were observed.

Rosar et al. [56] assessed the molecular imaging response by [^68^Ga]Ga-PSMA-11 uptake in total tumor burden and biochemical response by serum PSA values. Seventeen patients with mCRPC were treated with one single cycle of [^225^Ac]Ac-PSMA-617/[^177^Lu]Lu-PSMA-617 tandem therapy. Both radioligands were administered simultaneously, with a mean activity of 4 MBq and 6 GBq for [^225^Ac]Ac-PSMA-617 and [^177^Lu]Lu-PSMA-617, respectively. Molecular imaging response was assessed with whole-body total lesion PSMA (TLP) and molecular tumor volume (MTV). PR according to PCWG3 criteria was observed in 5 (29%) out of 17 patients with a TLP/MTV decline of >30%. SD was observed in 7 (41%) out of 17 patients with changes in TLP and MTV between −30% and +30%. PD was observed in 4 (24%) out of 17 patients with a TLP/MTV increase >30%. One patient with changes in TLP and MTV between −30% and +30% showed new metastases and was classified as PD. In addition, PSA decline >50% was observed in 5 (29%) out of 17 patients, whereas PSA increase >25% was observed in 4 (24%) out of 17 patients. Overall, tandem RLT was well-tolerated without any serious adverse events. Only one patient experienced grade 1 xerostomia. The median PFS and OS were 3.7 and 8.3 months, respectively.

Although both studies showed promising results, [^225^Ac]Ac-PSMA-617/[^177^Lu]Lu-PSMA-617 tandem therapy needs to be further investigated. In both studies, patients were only treated with one cycle of [^225^Ac]Ac-PSMA-617/[^177^Lu]Lu-PSMA-617 tandem therapy. The therapy effect may be increased when more cycles are added. However, toxicity may also increase. Therefore, future studies should focus on the trade-off between effectivity and safety when adding more cycles.

On the side note, it is important to mention that some patients, who were treated with either ^177^Lu-PSMA or ^225^Ac-PSMA or tandem therapy, do not show any treatment response at all due to tumors without PSMA expression or due to neuroendocrine differentiation. Neuroendocrine differentiation in PCa is mostly therapy-induced (17–30%); however, in some cases it develops de novo (0.5–2%) [58].

## 5. Future Applications

### 5.1. ^177^Lu-PSMA in Earlier Stages of PCa

After the Food and Drug Administration (FDA) granted approval for ^177^Lu-PSMA-617 (PLUVICTO) on 23 March 2022, adult patients with PSMA-positive mCRPC (who have been previously treated with androgen receptor inhibitors and taxane-based chemotherapy) can be treated with ^177^Lu-PSMA-617 in the USA [59]. In addition, a recent systematic review showed that ^177^Lu-PSMA RNT is highly effective and safe as a third-line treatment in patients with mCRPC [60]. Up until recently, ^177^Lu-PSMA RNT has only been used for the treatment of patients with mCRPC, but efficacy is also considered in patients with oligometastatic hormone-sensitive PC (oHSPC). In patients with oHSPC, increasing evidence shows a better outcome of metastases-directed therapy (MDT), such as external beam radiotherapy or targeted surgery. Therefore, Privé et al. [61] designed a phase II trial and hypothesized that ^177^Lu-PSMA RNT is also effective in patients with oHSPC, thereby postponing the need for ADT. Fifty-eight patients will be included in this two-arm randomized open-label multicenter phase II study comparing [^177^Lu]Lu-PSMA-I&T with the current standard of care (watchful waiting until initiation of ADT). The primary objective is to study the effect of [^177^Lu]Lu-PSMA-I&T in patients with oHSPC. In the treatment arm, patients will receive two cycles of [^177^Lu]Lu-PSMA-I&T with a dose of 7.4 GBq each. Depending on the results of this study, [^177^Lu]Lu-PSMA-I&T could be applied in earlier stages of prostate cancer.

Currently, other phase III trials (such as PSMAfore (NCT04689828) and PSMAddition (NCT04720157) study) are still ongoing, which are investigating therapeutic effects of ^177^Lu-PSMA RNT compared to other treatment options in mCRPC setting, but also in the earlier metastatic hormone-sensitive prostate cancer (mHSPC) setting.

### 5.2. ^225^Ac-PSMA

Xerostomia is the most common side effect seen in patients treated with ^225^Ac-PSMA RNT. Xerostomia is often the main reason to discontinue treatment. Therefore, strategies to reduce salivary toxicity are needed. Sialendoscopy with dilatation, saline irrigation and steroids (prednisolone) has been investigated in patients with mCRPC who were treated with ^225^Ac-PSMA. Rathke et al. [62] reported their first clinical experience using this method. Eleven patients with radiation sialadenitis in both parotid and submandibular glands received sialendoscopy before and after ^225^Ac-PSMA RNT. Health-related quality of life (HRQOL) was evaluated using the xerostomia questionnaire (XQ) and xerostomia inventory (XI). None of the patients experienced complications after sialendoscopy, and a significant improvement in HRQOL was observed. In another study, Belli et al. [63] measured the efficacy of salivary gland protectors in patients receiving ^177^Lu-PSMA RNT and performed a predictive dosimetry evaluation of ^225^Ac-PSMA RNT to assess the impact of salivary gland protectors in RNT. Salivary gland protectors include cooling with ice packs and folic glutamate tablets. Thirteen patients were included for the assessment of the predictive dosimetry evaluation. A reduction in absorbed dose was observed with the ^225^Ac-PSMA predictive dosimetry. The predicted dose of ^225^Ac-PSMA in salivary glands was decreased by 53% compared to the study population in the study of Kratochwil et al. [18] (1.08 vs. 2.33 Sv_RBE5_/MBq, respectively). Using the external beam radiotherapy biological model, the incidence of grade 2 or higher xerostomia was predicted. Without the salivary gland protectors, the predicted probability values were 97% with a dose of 50 kBq/kg and 100% with a dose of 100, 150 or 200 kBq/kg. With the salivary gland protectors, the predicted probability values were 40% with a dose of 50 kBq/kg, 94% with a dose of 100 kBq/kg and 100% with a dose of 150 and 200 kBq/kg. The results of this study suggest a possibility of reducing salivary gland toxicity with salivary gland protectors; however, these results need to be confirmed in future prospective studies with toxicity endpoints and dosimetry procedures.

Despite the literature regarding ^225^Ac-PSMA treatment, a formal phase I dose-escalation study has never been performed. One of the reasons is the limited production capacity of Ac-225 for therapeutic use [64], and there is also a lack of standardized techniques for the good manufacturing practice (GMP) production of [^225^Ac]Ac-PSMA. A recent publication reported on the production of [^225^Ac]Ac-PSMA-I&T according to GMP guidelines for the treatment of mCRPC [48]. In this study, high-quality [^225^Ac]Ac-PSMA-I&T was achieved with a radiochemical yield of >95% and a radiochemical purity of >90% (up to 3 h). To date, the ^225^Ac-PSMA used for the treatment of patients with mCRPC was not GMP-produced, and the quality is therefore debatable. With the future perspective that ^225^Ac-PSMA therapy may be applied in the daily clinical practice, a standardized way to produce high-quality ^225^Ac-PSMA is absolute necessary, as this may have an impact on the outcomes for the patients. Currently, two phase I studies are running at the moment: one with ^225^Ac-PSMA-617 (AAA/Novartis, NCT04597411) and one with ^225^Ac-PSMA I&T. (Figure 1 shows an example of partial response in a patient treated in this study).

### 5.3. Availability and Production of Lu-177 and Ac-225

Lu-177 was, until recently, used most widely for the production of ^177^Lu-DOTATATE for the treatment of neuroendocrine tumors [65]. However, after the favorable results of the VISION trial [16], there is a rise in demand for Lu-177. Currently, over 90% of the Lu-177 produced worldwide is used for both ^177^Lu-PSMA and ^177^Lu-DOTATATE. The most common way to largely produce Lu-177 is by using a nuclear reactor. The production can be achieved either directly using Lu-176 or indirectly by irradiating enriched ytterbium-176. A less common way is the cyclotron production route; however, it is still unsuitable to meet clinical demand due to low batch yields [65].

Ac-225 has an even more limited production compared to Lu-177. Currently, production of Ac-225 still relies on the extraction of Ac-225 from thorium-229, and with only three sources (Institute for Transuranium Elements (Karlsruhe, Germany), Oak Ridge National Laboratory (Oak Ridge, TN, USA) and Institute of Physics and Power Engineering (Obninsk, Russia)) worldwide, the supply is still limited. Newer techniques with accelerator-based routes (such as spallation of thorium-232, irradiation of radium-226 with protons or deuterons or irradiation of radium-226 based on photonuclear reaction) have been investigated to increase the production of Ac-225; however, each technique still has limitations and needs to be further developed. Ultimately, criteria for the quality of Ac-225 produced with the accelerator-based routes need to be established before it can be safely implemented in the daily clinical routine.

As it stands now, with the rising demand and the limited production of both Lu-177 and Ac-225, ^177^Lu-PSMA and ^225^Ac-PSMA RNT will not be able to replace the classic treatments for mCRPC. Therefore, there is a need for other alternatives.

### 5.4. Promising Radionuclides beyond Lu-177 and Ac-225

Besides Lu-177 and Ac-225, there are a few other alpha- and beta-emitters which are suitable for therapeutic applications, such as astatine-211 (At-211), bismuth-212 (Bi-212), bismuth-213 (Bi-213), lead-212 (Pb-212), radium-223 (Ra-223), terbium-149 (Tb-149), terbium-161 (Tb-161) and thorium-227 (Th-227). In 2021, an overview of the availability, chemical design and the preclinical and clinical data of these radionuclides has been summarized by Eychenne et al. [66]. As applications for therapy of metastatic PCa, Bi-213, Ra-223 and Th-227 have found their way in the (early) clinical setting. Other radionuclides such as At-211, Bi-212, Pb-212, Tb-149 and Tb-161 are still in development.

Bi-213 is an alpha-emitter with a short half-life of 46 min. Until today, only one mCRPC patient have been treated with Bi-213-PSMA-617 [67]. This patient received two cycles of [^213^Bi]Bi-PSMA-617 with a cumulative dose of 592 MBq. Remarkable molecular imaging response using [^68^Ga]Ga-PSMA PET/CT was observed after 11 months with a PSA decrease of 82%. Radiation dosimetry of Bi-213-PSMA-617 has been estimated in three mCRPC patients who all underwent PET scans with [^68^Ga]Ga-PSMA-617 [68]. Ga-68 was extrapolated to the half-life of Bi-213. A relative biological effectiveness of 5 was assumed for the alpha radiation. In conclusion, [^213^Bi]Bi-PSMA-617 was found to be sufficient for clinical application. However, compared to [^225^Ac]Ac-PSMA-617, the therapeutic index is clearly inferior and therefore a second choice.

Both Tb-149 and Tb-161 are suitable for therapy and have been explored preclinically. The difference between both radionuclides is that Tb-149 emits alpha particles and Tb-161 emits beta particles. Recently, [^149^Tb]Tb-PSMA-617 has been investigated in a preclinical setting in mice with PSMA-positive PCa tumors [69]. Dosimetry estimations showed a mean absorbed tumor dose of 6.9 Sv_RBE5_/MBq for [^149^Tb]Tb-PSMA-617, which was twice the calculated value compared to [^177^Lu]Lu-PSMA-617 (3.2 gray/MBq). However, the mean absorbed dose of the kidneys after [^149^Tb]Tb-PSMA-617 was ten times higher than [^177^Lu]Lu-PSMA-617. Nevertheless, it is expected that the safety threshold dose of 23 gray will not be exceeded with [^149^Tb]Tb-PSMA-617 since the required activity will likely be lower. Thus far, [^225^Ac]Ac–PSMA-617 shows the most favorable results and outperforms [^213^Bi]Bi-PSMA-617, as earlier mentioned, although Tb-149 seems more favorable in reducing the off-target dose since Tb-149 does not have relevant daughters emitting additional alpha particles. The main limitation of Tb-149 is the availability due to the specific technology that is needed for its production. For Tb-161, preclinical data are also available. One study [70] investigated the application of [^161^Tb]Tb-PSMA-617 for RNT using mice with PSMA-positive PCa tumors. Both in vitro and in vivo results were superior of [^161^Tb]Tb-PSMA-617 compared to [^177^Lu]Lu-PSMA-617. Furthermore, an activity-dependent increase in the median survival of 36 and 65 days was found with doses of 5 MBq/mouse and 10 MBq/mouse, respectively (compared to 19 days for the control group). Recently, dosimetry analysis for [^161^Tb]Tb-PSMA has also been performed [71]. The potential of [^161^Tb]Tb-PSMA to achieve a high probability of metastatic control in different metastatic scenarios was described using different models. The results showed that the required absorbed dose for metastatic control of [^161^Tb]Tb-PSMA (210–280 gray) was less than 40% compared to [^177^Lu]Lu-PSMA-617 (560–780 gray). In conclusion, [^161^Tb]Tb-PSMA seems to have the highest potential for improving the response rate of advanced metastatic PCa based on the dosimetry models used in this study.

Pb-212 is actually a beta-emitter with a half-life of 10.6 h, but alpha particles are mostly produced via the short-lived daughters Bi-212 and polonium-212 (Po-212) [72]. Preclinical analyses with [^212^Pb]Pb-PSMA-617 have only been performed by Stenberg et al. [72,73]. In vitro, [^212^Pb]Pb-PSMA-617 showed high affinity to C4-2 cells with high binding and internalization. In vivo cell binding ability of [^212^Pb]Pb-PSMA-617 was also tested in mice bearing human prostate C4-2 xenografts and showed a cell-binding fraction between 45 and 65%. Biodistribution data showed the highest activities in the urine, kidneys and tumors, with a tumor-to-kidney ratio between 0.25 and 0.34. These results are very promising; however, more preclinical data need to be collected to validate the potential therapeutic effects of [^212^Pb]Pb-PSMA-617.

Lastly, Th-227 has also emerged as a promising candidate. Th-227 is an alpha emitter with a half-life of 18.7 days [66]. For PCa, only preclinical data are available for the compound PSMA-targeted thorium-227 conjugate (PSMA TTC, BAY2315497) [74]. In vitro analysis showed a rapid increase in intracellular radioactivity after incubation of PSMA-TTC with PSMA-high-expressing cells, whereas PSMA-negative cells showed no uptake. Furthermore, a dose-dependent antitumor effect was found in androgen-responsive PCa cells using 75, 150 and 300 kBq/kg with a maximum antibody dose of 0.43 mg/kg. More advanced stages of PCa were also mimicked in patient-derived models to test PSMA TTC, which showed similar antitumor effects. The data presented by this study show the potential of PSMA TTC, and a phase I study has been initiated (NCT03724747) using PSMA TTC in patients with mCRPC.

Even though alpha-emitters offer advantages (high LET and limited range in healthy tissue) over beta-emitters, the recoil effect of alpha-emitters needs to be taken into consideration. The recoil effect of an alpha-emitter leads to the release of their radioactive daughters, which may lead to the irradiation of healthy tissues, which could cause severe radiotoxic effects such as organ dysfunction [75]. Characteristics of the alpha radiopharmaceuticals, such as biological half-life, carrier stability or elimination route, may play a vital role in the severity of the radiotoxic effects [75]. The recoiling daughters are distributed through the body in three different manners: firstly, by the distance covered due to recoil energy; secondly, by diffusion processes; and lastly, by active transport, such as blood flow [76]. Depending on the location of the recoiled daughters, each of the distribution processes may contribute to the severity of the radiotoxic effects.

Ultimately, each of the mentioned radionuclides showed promising results in either the preclinical or clinical setting, or both. However, challenges still remain in the production process of most of these radionuclides. In addition, potential long-term risks of alpha-emitters are still not fully known due to the lack of long-term data from clinical studies. Nonetheless, all mentioned alpha- and beta-emitters are still in development and may provide new therapeutic options in the future.

## 6. Conclusions

After FDA approval on 23 March 2022, treatment with ^177^Lu-PSMA-617 is already possible for adult patients with PSMA-positive mCRPC in the USA. Although promising results have been achieved with ^225^Ac-PSMA RNT, further developments in this field are needed before ^225^Ac-PSMA RNT can be implemented in daily clinical practice. Clinical developments, such as comparing PSMA RNT with the standard of care in larger groups or applying these therapies in earlier stages of prostate cancer, are surely important. However, future clinical studies should not only focus on comparing PSMA RNT with different standard treatments in different treatment settings, but also on minimizing the toxicity profile (especially xerostomia) of RNT. Furthermore, to meet the rising future demand of ^225^Ac-PSMA RNT, accelerator-based production of Ac-225 needs to be further developed. Finally, the exploration of other potential radionuclides is inevitable, as alternative therapeutic options are urgently needed in the future.

## Figures and Tables

**Figure 1 pharmaceutics-14-02166-f001:**
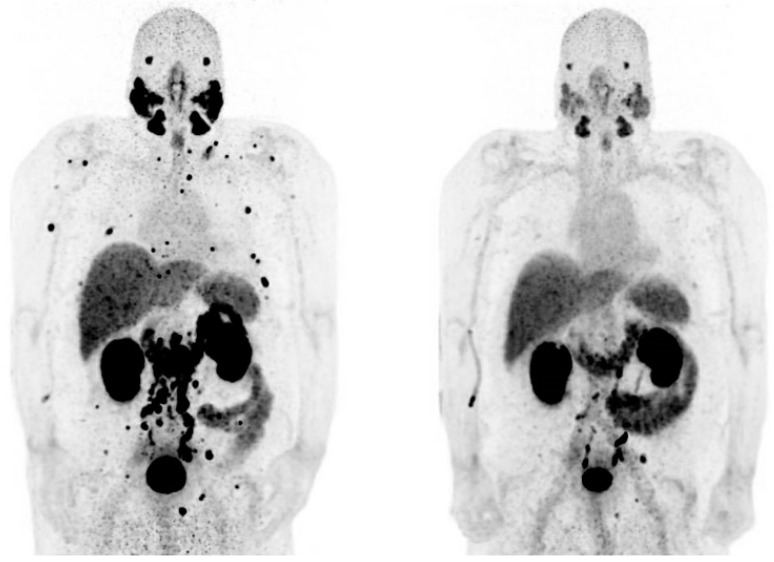
^68^Ga-PSMA I&T PET demonstrating a partial response after 2 cycles of ^225^Ac-PSMA I&T in a patient with bone-only mCRPC. Left = before ^225^Ac-PSMA I&T therapy. Right = after ^225^Ac-PSMA I&T therapy. ^225^Ac = actinium-225. ^68^Ga = gallium-68. I&T = Imaging and Therapy. mCRPC = metastatic castration-resistant prostate cancer. PET = positron emission tomography. PSMA = prostate-specific membrane antigen.

**Table 1 pharmaceutics-14-02166-t001:** Summary of 177Lu-PSMA studies.

Author	N	Number of Cycles	Injected Activity (GBq)	PSA Response (%)	Radiological Response (%)	PFS and/or rPFS(mo)	OS and/or rOS (mo)
**Heck et al. (2016) [6]**	19	40 (total in study)	7.3 per cycle	≥30% in 10/19 (53) ≥50% in 6/19 (32) ≥90% in 2/19 (11)	NR	4.1 (PFS)	12.9 (OS)
**Kesavan et al. (2018) [8]**	20	Median 4 per patient	5.5 per cycle	≥50% in 8/20 (40)	NR	NR	NR
**Heck et al. (2019) [9]**	100	319 (total in study)	7.4 per cycle	≥30% in 47/100 (47) ≥50% in 38/100 (38) ≥90% in 11/100 (11)	NR	NR	NR
**Barber et al. (2019) [10]**	131	Median 3 per patient	6.3 per cycle	Any in 25/62 T-pretreated (40) Any in 40/70 T-naïve (57)	NR	6 in T-pretreated (rPFS) 8.8 in T-naïve (rPFS)	10.7 in T-pretreated (OS) 27.1 in T-naïve (OS)
**Von Eyben et al. (2019) [11]**	45	Median 3 per patient	Cumulative median 14.5	>50% in 36/45 (80) >90% in 25/45 (56)	NR	16 (PFS) 18 (rPFS)	NR
**Suman et al. (2019) [12]**	40	Median 3 per patient	4.4–5.5 per cycle	CR (>50%) in 17/40 (43) PR (30–50%) in 2/40 (5) SD (<30%) in 2/40 (5) PD (any increase) in 19/40 (48)	CR in 2/40 (5) PR in 5/40 (13) SD in 9/40 (23) PD in 21/40 (53)	NR	NR
**Gallyamov (2020) [13]**	110	327 (total in study)	6.34 per cycle (mean)	>50% in 60/110 (55) <50% or failure in 38/110 (35)	NR	NR	NR
**Bulbul et al. (2020) [14]**	45	164 (total in study)	7.4 per cycle	≥50% in 15/45 (33) ≥25% in 20/45 (44)	NR	7.4	17.1 (OS)
**Hofman (2021) [15]**	89	Median 5 per patient	6–8.5 per cycle	>50% in 66%	NR	5.1	NR
**Sartor (2021) [16]**	831	Maximum 6 per patient	7.4 per cycle	≥50% in 177/385 (46) ≥80% in 127/385 (33)	CR in 17/184 (9) PR in 77/184 (42)	8.7 (rPFS)	15.3 (OS)
**Rosar (2022) [42]**	66	2 per patient	7.1 per cycle (mean)	PR in 34/66 (51.5) SD in 20/66 (30.3) PD in 12/66 (18.2)	PR in 40/66 (60.6) SD in 19/66 (28.7) PD in 7/66 (10.6)	NR	18 (OS)

GBq = gigabecquerel. PSA = prostate-specific antigen. N = number of patients. PFS = progression-free survival (both clinical and radiological). rPFS = radiographic PFS. OS = overall survival. Mo = months. NR = not reported. CR = complete remission. PR = partial remission. SD = stable disease. PD = progression disease. T-pretreated = taxane chemotherapy-pretreated. T-naïve = Taxane-naïve.

**Table 2 pharmaceutics-14-02166-t002:** Frequencies of toxicities with ^177^Lu-PSMA and ^225^Ac-PSMA RNT according to Common Terminology Criteria for Adverse Events (CTCAE) version 5.0.

Toxicity	Radionuclide	Studies	Any Grade Toxicity	Grade 3 or Higher Toxicity
		Number of studies (ref)	Number of patients (%)	Number of patients (%)
**Xerostomia**	^177^Lu	6 [7,8,9,12,15,16]	299 out of 806 (37)	NR
	^225^Ac	10 [17,18,20,21,22,23,24,25,26,27]	161 out of 210 (77)	2 out of 12 (17)
**Xeropthalmia**	^177^Lu	1 [15]	29 out of 98 (30)	NR
	^225^Ac	2 [18,21]	5 out of 87 (6)	NR
**Dysgeusia**	^177^Lu	1 [15]	12 out of 98 (12)	NR
	^225^Ac	2 [21,22]	10 out of 87 (11)	NR
**Fatigue**	^177^Lu	4 [7,9,15,16]	358 out of 746 (48)	36 out of 627 (6)
	^225^Ac	6 [18,21,22,23,24,26]	73 out of 166 (44)	1 out of 28 (4)
**Anorexia**	^177^Lu	3 [7,9,16]	137 out of 648 (21)	10 out of 529 (2)
	^225^Ac	4 [21,22,24,26]	36 out of 124 (29)	NR
**Diarrhea**	^177^Lu	3 [9,15,16]	130 out of 727 (18)	5 out of 627 (0.8)
	^225^Ac	1 [26]	1 out of 11 (9)	NR
**Obstipation**	^177^Lu	2 [7,16]	115 out of 548 (21)	6 out of 529 (1.1)
	^225^Ac	2 [21,26]	21 out of 84 (25)	NR
**Nausea**	^177^Lu	4 [8,9,15,16]	239 out of 747 (32)	8 out of 627 (1.3)
	^225^Ac	4 [18,21,22,26]	23 out of 112 (21)	NR
**Vomiting**	^177^Lu	2 [15,16]	117 out of 627 (19)	6 out of 627 (1)
	^225^Ac	2 [21,26]	5 out of 84 (6)	NR
**Hematological toxicity ***	^177^Lu	1 [12]	7 out of 40 (18)	NR
	^225^Ac	NR	NR	NR
**Anemia**	^177^Lu	5 [7,8,9,10,16]	302 out of 835 (36)	84 out of 816 (10)
	^225^Ac	7 [17,20,21,22,23,24,26]	52 out of 170 (31)	18 out of 169 (11)
**Leukopenia**	^177^Lu	3 [10,15,16]	119 out of 794 (15)	16 out of 794 (2)
	^225^Ac	6 [17,18,21,22,24,26]	26 out of 139 (19)	10 out of 113 (9)
**Neutropenia**	^177^Lu	4 [7,8,9,15]	42 out of 237 (18)	13 out of 218 (6)
	^225^Ac	NR	NR	NR
**Thrombocytopenia**	^177^Lu	6 [7,8,9,10,15,16]	230 out of 933 (25)	61 out of 914 (7)
	^225^Ac	7 [17,18,21,22,24,26,27]	22 out of 151 (15)	9 out of 122 (7)
**Renal toxicity**	^177^Lu	1 [10]	16 out of 167 (10)	1 out of 167 (0.6)
	^225^Ac	4 [21,22,23,26]	27 out of 126 (21)	6 out of 84 (7)

RNT = radionuclide therapy. NR = not reported. * = not specified. Ref = reference. CTCAE = Common Terminology Criteria for Adverse Events.

**Table 3 pharmaceutics-14-02166-t003:** Summary of ^225^Ac-PSMA studies.

Author	N	Number of Cycles	Injected Activity (MBq)	PSA Response	Radiological Response	Median PFS (mo)	Median OS (mo)
**Kratochwil (2017) [18]**	14	NR	50, 100, 150 or 200 kBq/kg	NR	NR	6	>12
**Kratochwil (2018) [19]**	40	3–5 per patient	100 kBq/kg per cycle	Any in 33/38 (87) >50% in 24/38 (63)	NR	NR	NR
**Sathekge et al. (2019) [20]**	17	59 (total in study)	7.4 per cycle	≥90% in 14/17 (82)	>50% in 15/17 (88)	NR	NR
**Sathekge et al. (2020) [21]**	73	210 (total in study)	8 (first cycle) 7, 6 or 4 (subsequent cycles)	Any in 60/73 (82) ≥50% in 51/73 (70)	NR	15.2	18
**Zacherl et al. (2020) [22]**	14	34 (total in study)	7.8 per cycle (mean)	Any in 11/14 (79) >50% in 7/14 (50)	NR	NR	NR
**Yadav et al. (2020) [23]**	28	85 (total in study)	100 kBq/kg per cycle	Any in 22/28 (79) >50% in 11/28 (39)	CR in 2/22 (9) PR in 10/22 (45) SD in 2/22 (9)	12	17
**Feuerecker et al. (2021) [24]**	26	61 (total in study)	9 per cycle (mean)	Any in 23/26 (88) ≥50% in 17/26 (65)	NR	4.1	7.7
**Van der Doelen (2020) [25]**	13	3 per patient (median)	8 (first cycle) 6 (subsequent cycles)	≥50% in 9/13 (69) ≥90% in 6/13 (46)	NR	NR	8.5
**Satapathy et al. (2020) [26]**	11	25 (total in study)	100 kBq/kg per cycle	≥50% in 5/11 (45)	NR	NR	NR
**Sanli et al. (2021) [27]**	12	25 (total in study)	7.4 per cycle (median)	After first cycle: Any in 9/12 (75) >50% in 6/12 (50) >85% in 5/12 (42)	CR in 4/10 (40) PR in 5/10 (50)	4	10

MBq = megabecquerel. kBq = kilobecquerel. PSA = prostate-specific antigen. N = number of patients. PFS= median progression-free survival. rPFS = radiographic PFS. OS = median overall survival. mo = months. NR = not reported. CR = complete remission. PR = partial remission. SD = stable disease.

## Data Availability

The data presented in this study are openly available in PubMed.

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
