# Peer review of "Advances in 177Lu-PSMA and 225Ac-PSMA Radionuclide Therapy for Metastatic Castration-Resistant Prostate Cancer"

_pharmaceutics, 2022, doi:10.3390/pharmaceutics14102166_

Round 1
Reviewer 1 Report (Previous Reviewer 1)
The paper and main conclusions are not updated and misleading. The major drawback that the authors do not follow this field and don’t know that on March 23, 2022, the FDA granted approval to Lu-177 vipivotide tetraxetan (Pluvicto; 177Lu-PSMA-617), for the treatment of patients with metastatic castration-resistant prostate cancer (mCRPC) in the post androgen receptor pathway inhibition, post-taxane-based chemotherapy setting.
Please see https://link.springer.com/article/10.1007/s40291-022-00594-2
Conclusions: "Although promising results have been achieved with 177Lu-PSMA RLT and 225Ac-PSMA TAT, further developments in this field are needed before these therapies can be implemented in the daily clinical practice"
Pluvicto (177Lu-PSMA RLT) is already in the daily clinical practice in the USA.
Author Response
We would like to express our gratitude for your consideration for publication and for the comments from the reviewers and editor. We have incorporated the constructive comments provided by the reviewers in order to improve our manuscript.
The original comments from the reviewers are listed below in italic, followed by our response to each comment. We have also submitted the revised manuscript with track changes which highlights the adjustments.
Reviewer 1
The paper and main conclusions are not updated and misleading. The major drawback that the authors do not follow this field and don’t know that on March 23, 2022, the FDA granted approval to Lu-177 vipivotide tetraxetan (Pluvicto; 177Lu-PSMA-617), for the treatment of patients with metastatic castration-resistant prostate cancer (mCRPC) in the post androgen receptor pathway inhibition, post-taxane-based chemotherapy setting.
Please see https://link.springer.com/article/10.1007/s40291-022-00594-2
Conclusions: "Although promising results have been achieved with 177Lu-PSMA RLT and 225Ac-PSMA TAT, further developments in this field are needed before these therapies can be implemented in the daily clinical practice"
Pluvicto (177Lu-PSMA RLT) is already in the daily clinical practice in the USA.
We thankfully agree with this important comment of the reviewer and have added the FDA approval under the section “Future applications” (line 362 tot 365). Also the conclusion have been rewritten with the addition of the FDA approval (lines 539-540).
Reviewer 2 Report (New Reviewer)
The topic of this review is highly actual and interesting to a wide audience. Majority of the previous reviewer´s comments were implemented in the manuscript, so I have just minor additional comments bellow:
Even though both RLT and TAT terms are commonly used in literature, I would suggest not to classify the treatment options as the radioligand therapy - RLT and the targeted alpha therapy - TAT, since both methods use the same peptide based radioligands and differ only in the radionuclide used and the radiations emitted. Please use alpha/beta radionuclide therapy or similar more systematic classification at least in the introduction part.
Please format Table 2 in a way that the second column is wider and the data are presented in better readable form. Possibly delete PSMA (always the same ligand) and add row separators between Ac/Lu.
It should be stressed and commented that in some cases there is no response nor for the 177Lu- nor for the 225Ac- PSMA therapy, nor combined therapy, e.g. due to neuroendocrine differentiation or non expressing of PSA.
The nuclear recoil effect in the decay of cascade alpha-emitters like 225Ac, 223Ra as well as 227Th should be pointed-out and the risks related to daughter radioactive progeny spread should be discussed in detail together with some references, since it could bring both positive and negative effects (diffusion within the tumor and irradiation of the whole tumor volume vs. non-targeted irradiation of healthy tissues). Also it should be noted that particularly for the alpha emitters, the long-term clinical studies data are missing and thus all the potential risks are not yet fully known even at the preclinical stages of the research.
Author Response
We would like to express our gratitude for your consideration for publication and for the comments from the reviewers and editor. We have incorporated the constructive comments provided by the reviewers in order to improve our manuscript.
The original comments from the reviewers are listed below in italic, followed by our response to each comment. We have also submitted the revised manuscript with track changes which highlights the adjustments.
Reviewer 2
The topic of this review is highly actual and interesting to a wide audience. Majority of the previous reviewer´s comments were implemented in the manuscript, so I have just minor additional comments bellow:
Even though both RLT and TAT terms are commonly used in literature, I would suggest not to classify the treatment options as the radioligand therapy - RLT and the targeted alpha therapy - TAT, since both methods use the same peptide based radioligands and differ only in the radionuclide used and the radiations emitted. Please use alpha/beta radionuclide therapy or similar more systematic classification at least in the introduction part.
We agree with this suggestion and have replace RLT and TAT with alpha and/or beta RNT (radionuclide therapy) throughout the manuscript (see track changes).
Please format Table 2 in a way that the second column is wider and the data are presented in better readable form. Possibly delete PSMA (always the same ligand) and add row separators between Ac/Lu.
We have removed PSMA, so therefore no need to make the second column wider. We also changed the lay-out with row separators between Ac and Lu as suggested for better visibility.
It should be stressed and commented that in some cases there is no response nor for the 177Lu- nor for the 225Ac- PSMA therapy, nor combined therapy, e.g. due to neuroendocrine differentiation or non expressing of PSA.
We fully agree with this statement and we have added that in some cases there is no response with a reference under the section “Combination therapy” (lines 354 to 358).
The nuclear recoil effect in the decay of cascade alpha-emitters like 225Ac, 223Ra as well as 227Th should be pointed-out and the risks related to daughter radioactive progeny spread should be discussed in detail together with some references, since it could bring both positive and negative effects (diffusion within the tumor and irradiation of the whole tumor volume vs. non-targeted irradiation of healthy tissues). Also it should be noted that particularly for the alpha emitters, the long-term clinical studies data are missing and thus all the potential risks are not yet fully known even at the preclinical stages of the research.
We agree with the reviewer regarding the nuclear recoil effect and also the lack of longterm studies for alpha emitters. Under the section “Promising radionuclides beyond Lu-177 and Ac-225” we have added a section on the nuclear recoil effect (lines 521 to 531).
Reviewer 3 Report (New Reviewer)
The study evaluated the advances in Lu PSAMA Radioligand therapy and AcPSMA targeted Alpha therapy for metastatic castration resistant prostate cancer. The authors have addressed all the issues more or less adequately. The manuscript is well organized and written. The topic is relevant and try to complete the knowledge gap that exist in the treatment of patients with metastatic castration resistant prostate cancer. I think the article should be accepted after minor revisions that could further improve this manuscript, which are the following 2 questions
1. The question that remain unanswered and should be clarified by the authors is that : Could Lu PSMA or Ac PSMA treatment replace the typical treatments in such patients? considering the limited production capacity of these radionuclides for therapeutic use?
2. The authors mentioned that the response rate after Lu PSMA treatment monotherapy is 30%. Which is the response rate after Ac PSMA monotherapy treatment, This should also be mentioned by the authors
Author Response
We would like to express our gratitude for your consideration for publication and for the comments from the reviewers and editor. We have incorporated the constructive comments provided by the reviewers in order to improve our manuscript.
The original comments from the reviewers are listed below in italic, followed by our response to each comment. We have also submitted the revised manuscript with track changes which highlights the adjustments.
Reviewer 3
The study evaluated the advances in Lu PSAMA Radioligand therapy and AcPSMA targeted Alpha therapy for metastatic castration resistant prostate cancer. The authors have addressed all the issues more or less adequately. The manuscript is well organized and written. The topic is relevant and try to complete the knowledge gap that exist in the treatment of patients with metastatic castration resistant prostate cancer. I think the article should be accepted after minor revisions that could further improve this manuscript, which are the following 2 questions
- The question that remain unanswered and should be clarified by the authors is that : Could Lu PSMA or Ac PSMA treatment replace the typical treatments in such patients? considering the limited production capacity of these radionuclides for therapeutic use?
Under the section “Availability and production of Lu-177 and Ac-225”, we have clarified that as it stands now Lu-PSMA and Ac-PSMA will not be able to replace classic treatments (lines 447 to 448).
- The authors mentioned that the response rate after Lu PSMA treatment monotherapy is 30%. Which is the response rate after Ac PSMA monotherapy treatment, This should also be mentioned by the authors
We agree with this statement of the reviewer. We have added the response rate after Ac-PSMA monotherapy under the section “Combination therapy” (lines 314 to 315).
Reviewer 4 Report (New Reviewer)
Review (Comments to the Author)
For the paper entitled: "Advances in 177Lu-PSMA Radioligand Therapy and 225Ac-PSMA Targeted Alpha Therapy for metastatic Castration-Resistant Prostate Cancer" are major and minor aspects that should be considered.
Due to the various major revisions by three reviewers before, the manuscript obviously had to be completely redesigned and rewritten. Unfortunately, since I do not have the previous version, I cannot adequately address some important points criticized by these reviewers and their probable implementation in the now amended manuscript. Therefore, in my view, a reassessment by the original reviewers is extremely important.
Introduction, Line 51: Please also emphasize here that in particular the aggressivetumor lesions of a prostate carcinoma show an increased PSMA expression and prove this with a reference.
Introduction, Line 53-54: "the most investigates PSMA ligands" -> please add here that this applies to therapy, but not to diagnostics.
Introduction: The references #28-29 and #31-32 (from 1997,1998, 1995and 1998) could be updated.
Introduction: In particular, with regard to the statement "There is also expression of PSMA in other tissues", current references should be given that prove that increased PSMA expression is also found in benign and malignant tumors that do not affect the prostate.
The term theranostics should be explained in the review. In particular, a study that compares 177Lu-PSMA with 225Ac-PSMA should show more the diagnostic pathway to therapy and discuss more the differences.
Line 312-313: Early molecular imaging response and its association with overall survival should be more discussed in the review (see a recent study from 2022from the European Journal of Nuclear Medicine and Molecular Imaging on this topic Rosar, Florian et al. “Early molecular imaging response assessment based on determination of total viable tumor burden in [68Ga]Ga-PSMA-11 PET/CT independently predicts overall survival in [177Lu]Lu-PSMA-617 radioligand therapy.” European journal of nuclear medicine and molecular imaging vol. 49,5 (2022): 1584-1594. doi:10.1007/s00259-021-05594-8.)
It is a pity that a review of prostate cancer therapy with 177Lu-PSMA or 225Ac-PSMA does not include any patient images (before and after the respective therapy).
With regard to the additional aspects of the future perspectives, the 177Lu-PSMA radioligand therapy should be discussed as a third-line treatment (see a review on this topic: von Eyben, Finn E et al. “177Lu-PSMA Radioligand Therapy Is Favorable as Third-Line Treatment of Patients with Metastatic Castration-Resistant Prostate Cancer. A Systematic Review and Network Meta-Analysis of Randomized Controlled Trials.” Biomedicines vol. 9,8 1042.19 Aug. 2021,doi:10.3390/biomedicines9081042).
The authors have put a lot of effort into the "Promising radionuclides beyond Lu-177 and Ac-225" chapter, which deserves praise.

Author Response
We would like to express our gratitude for your consideration for publication and for the comments from the reviewers and editor. We have incorporated the constructive comments provided by the reviewers in order to improve our manuscript.
The original comments from the reviewers are listed below in italic, followed by our response to each comment. We have also submitted the revised manuscript with track changes which highlights the adjustments.
Reviewer 4
For the paper entitled: "Advances in 177Lu-PSMA Radioligand Therapy and 225Ac-PSMA Targeted Alpha Therapy for metastatic Castration-Resistant Prostate Cancer" are major and minor aspects that should be considered.
Due to the various major revisions by three reviewers before, the manuscript obviously had to be completely redesigned and rewritten. Unfortunately, since I do not have the previous version, I cannot adequately address some important points criticized by these reviewers and their probable implementation in the now amended manuscript. Therefore, in my view, a reassessment by the original reviewers is extremely important.
Introduction, Line 51: Please also emphasize here that in particular the aggressivetumor lesions of a prostate carcinoma show an increased PSMA expression and prove this with a reference.
We have added the emphasis on the increased PSMA expression of aggressive tumors and also proved it with a reference (line 52 to 53)
Introduction, Line 53-54: "the most investigates PSMA ligands" -> please add here that this applies to therapy, but not to diagnostics.
We have added that it applies to therapy as suggested (line 60)
Introduction: The references #28-29 and #31-32 (from 1997,1998, 1995and 1998) could be updated.
We have update the references as suggested (lines 48 and 50) with:
- Yao V, Berkman CE, Choi JK, O'Keefe DS, Bacich DJ. Expression of prostate-specific membrane antigen (PSMA), increases cell folate uptake and proliferation and suggests a novel role for PSMA in the uptake of the non-polyglutamated folate, folic acid. Prostate. 2010;70(3):305-16.
- Mhawech-Fauceglia P, Zhang S, Terracciano L, Sauter G, Chadhuri A, Herrmann FR, et al. Prostate-specific membrane antigen (PSMA) protein expression in normal and neoplastic tissues and its sensitivity and specificity in prostate adenocarcinoma: an immunohistochemical study using mutiple tumour tissue microarray technique. Histopathology. 2007;50(4):472-83.
- Nishida H, Kondo Y, Kusaba T, Kadowaki H, Daa T. Immunohistochemical Reactivity of Prostate-Specific Membrane Antigen in Salivary Gland Tumors. Head Neck Pathol. 2022;16(2):427-33.
Introduction: In particular, with regard to the statement "There is also expression of PSMA in other tissues", current references should be given that prove that increased PSMA expression is also found in benign and malignant tumors that do not affect the prostate.
We have added new references with regarding to this comment:
- Mhawech-Fauceglia P, Zhang S, Terracciano L, Sauter G, Chadhuri A, Herrmann FR, et al. Prostate-specific membrane antigen (PSMA) protein expression in normal and neoplastic tissues and its sensitivity and specificity in prostate adenocarcinoma: an immunohistochemical study using mutiple tumour tissue microarray technique. Histopathology. 2007;50(4):472-83.
- Nishida H, Kondo Y, Kusaba T, Kadowaki H, Daa T. Immunohistochemical Reactivity of Prostate-Specific Membrane Antigen in Salivary Gland Tumors. Head Neck Pathol. 2022;16(2):427-33.
The term theranostics should be explained in the review. In particular, a study that compares 177Lu-PSMA with 225Ac-PSMA should show more the diagnostic pathway to therapy and discuss more the differences.
We agree with this statement of the reviewer and added an explanation regarding theranostics in the introduction (lines 55 tot 57)
Line 312-313: Early molecular imaging response and its association with overall survival should be more discussed in the review (see a recent study from 2022from the European Journal of Nuclear Medicine and Molecular Imaging on this topic Rosar, Florian et al. “Early molecular imaging response assessment based on determination of total viable tumor burden in [68Ga]Ga-PSMA-11 PET/CT independently predicts overall survival in [177Lu]Lu-PSMA-617 radioligand therapy.” European journal of nuclear medicine and molecular imaging vol. 49,5 (2022): 1584-1594. doi:10.1007/s00259-021-05594-8.)
We agree that early imaging response and its association with the OS should be more discussed in this manuscript. We have added a paragraph with the suggested reference (lines 195 to 201) and also added the data of the suggested reference to Table 1.
It is a pity that a review of prostate cancer therapy with 177Lu-PSMA or 225Ac-PSMA does not include any patient images (before and after the respective therapy).
We thank the reviewer for this comment and we have added an image of one of our patient who were treated with 225Ac-PSMA I&T, which shows radiographic response (lines 450 to 453).
With regard to the additional aspects of the future perspectives, the 177Lu-PSMA radioligand therapy should be discussed as a third-line treatment (see a review on this topic: von Eyben, Finn E et al. “177Lu-PSMA Radioligand Therapy Is Favorable as Third-Line Treatment of Patients with Metastatic Castration-Resistant Prostate Cancer. A Systematic Review and Network Meta-Analysis of Randomized Controlled Trials.” Biomedicines vol. 9,8 1042.19 Aug. 2021,doi:10.3390/biomedicines9081042).
We agree with the comment regarding Lu-PSMA therapy as a third line therapy. We have discussed this point with the suggested reference under the section “Lu-PSMA in earlier stages of PCa” (lines 365 to 367).
The authors have put a lot of effort into the "Promising radionuclides beyond Lu-177 and Ac-225" chapter, which deserves praise.
We thank the reviewer for the comment.
Reviewer 5 Report (New Reviewer)
- As the Journal calls, 'Pharmaceutics', Introduction-paragraph needed to be further detailed or worked out, f.i. what means PSMA-617 and PSMA-I&T ? (as done by Hartrampf in Eur J Nucl Mol Imaging, 2022, 49, 3269-76) and editing the molecular structures of the 2 ligands, as well maybe for the *Ga-PSMA-11 compound - makes the manuscript more attractive,
- Maybe, some tables can be better presented (or other layout)
- Conclusions : can the 2 ligands be compared to each other regarding effectiveness, as you're using the 2 ligands for treatment of the same disorder ?! I am missing this !
Author Response
We would like to express our gratitude for your consideration for publication and for the comments from the reviewers and editor. We have incorporated the constructive comments provided by the reviewers in order to improve our manuscript.
The original comments from the reviewers are listed below in italic, followed by our response to each comment. We have also submitted the revised manuscript with track changes which highlights the adjustments.
Reviewer 5
- As the Journal calls, 'Pharmaceutics', Introduction-paragraph needed to be further detailed or worked out, f.i. what means PSMA-617 and PSMA-I&T ? (as done by Hartrampf in Eur J Nucl Mol Imaging, 2022, 49, 3269-76) and editing the molecular structures of the 2 ligands, as well maybe for the *Ga-PSMA-11 compound - makes the manuscript more attractive,
We thank the reviewer for this comment and have added an explanation on the utility of PSMA-11, PSMA-617 and PSMA I&T (either diagnostic or therapeutic) (lines 58-60).
Due to copyrights we could not use existing figures of the molecular structures of the different PSMA. Furthermore, unfortunately we do not have the resources to edit the structures ourselves.
- Maybe, some tables can be better presented (or other layout)
We have changed the lay-out of table 1, 2 and 3 for better visibility.
- Conclusions : can the 2 ligands be compared to each other regarding effectiveness, as you're using the 2 ligands for treatment of the same disorder ?! I am missing this !
Regarding the comparability of the 2 ligands, we have added a sentence under the section “Introduction” (line 60 to 61) which state that both PSMA-617 and PSMA-I&T shows comparable behavior in-vivo, toxicity profile and effectivity in patients with mCRPC. References are added too.
Round 2
Reviewer 1 Report (Previous Reviewer 1)
I have no additional comments.
Reviewer 4 Report (New Reviewer)
no further comments
This manuscript is a resubmission of an earlier submission. The following is a list of the peer review reports and author responses from that submission.
Round 1
Reviewer 1 Report
The main goal of the present review was to summarize the clinical data of 177Lu-PSMA and 225Ac-PSMA therapies in patients with mCRPC and provide the future perspective on RLT. A few similar reviewers have been published earlier. The review is very superficial and not systematic, the novelty is missing. The newest important papers are not discussed. The criteria for the selection of patients for radioligand therapies are not described. It seems that the authors have copied many sentences from other papers, made slight changes and created the present manuscript. Comparison of PSMA-617 PSMA I&T ligands has not been performed.
Abstract
Many other drugs, not only “classic treatment options with chemotherapy and drugs targeting androgen signaling” are uses for mCRPC treatment (sipuleucel-T and pembrolizumab (immunotherapy), olaparib and rucaparib (PARP inhibitors) and 223Ra).
Introduction
The first paragraph in the introduction must be updated.
Line 16: “PSMA) receptors, which are upregulated up to thousand times more in prostate cancer cells compared to the cells in normal tissues.”
Lines 48-49: “However, the expression on prostate cancer cells is a thousand-fold higher than expression on normal tissues (7).
Please add a good reference showing the “real numbers” of PSMA expression in different tissues, including kidneys and salivary glands, not the reference 7.
However, not all patients with mCRPC have PSMA expression. At the same time, “a high PSMA imaging score on SPECT or PET/CT is an independent prognostic indicator of poor OS in mCRPC, which is concordant with PSMA’s biological role as a hallmark of lethal, aggressive mCRPC” [PubMed PMID: 33680970].
Lines 55-57: “Lu-177 emits gamma rays on decay with most abundant peaks of 112.9 and 208.4 keV. These gamma rays enable image acquisition and dosimetry calculation using a gamma camera (8-10)”.
I have opened all three references 8-10, and was not able to find any SPECT-CT scan of 177Lu-PSMA. In all cites papers PET-CT imaging with 68Ga-PSMA-11 or 18F-PSMA-1007 used. I don’t have time to check all references.
Theranostic concept for mCRPC expressing prostate specific membrane antigens (PSMA) is not discussed (68Ga-PSMA-11 (FDA, 2020), 18F-DCFPyl (FDA, 2021)).
Lines 61-62: “The cascade also includes 2 beta disintegrations of 1.6 and 0.6 MeV and gamma co-emissions are generated from disintegrations of Francium-221 and Bismuth-213, which may be used for imaging (11).”
Theoretically yes, but practically it is not possible: high doses (toxic) of 225Ac-PSMA must be used for such imaging.
Lines 80-82: “Therapy was well tolerated in one patient without significant hematological or renal toxicity and no abnormalities were found in other laboratory parameters.”
Was badly tolerated in the second patient?
Reviewer 2 Report
The authors present an interesting and complete literature review regarding recent clinical studies performed with Lu-177 and Ac-225 for the treatment of patients with prostate cancer. The references cited include relevant work on targeted radionuclide therapy. The flow of the review is appropriate, and the structure is well organized. I have a couple of major points to consider based on the current version of this review article:
- There is a gap between the title and the abstract since key advances in RLT and TAT are not described in the latter. The abstract shows a brief overview of how mCRPC is treated and a couple of general reasons regarding the use of PSMA as targeting moiety. This gap is also reflected in the body of the manuscript, where the authors present the main points of different clinical studies without highlighting what the “advances” in RLT and TAT have been.
- Why would be a reader interested in this article? The abstract does not provide enough information to motivate the reader and does not highlight what differentiates this work with respect to other reviews related with the topic. The authors referred multiple times to systematic reviews on both Lu-177 and Ac-225, which brings an important question into this review. What is the added value of this review article with respect to those systemic reviews? Are the authors actually doing a comprehensive and “new/novel” discussion of the advances made for treatment of mCRPC?
- The review article will benefit from graphical representations. Authors should consider using plots to compare the response to treatment based on number of cycles and activity per cycle.
- What is the input/perspective of the authors regarding the tandem use of RLT and TAT? Is one cycle of RLT+TAT a good representation of the treatment and response with monotherapies?
Overall, this review article is missing the perspective and input of the authors regarding the advances of RLT and TAT. Consider including a perspective section to complement the summary of results that were presented. Additionally, the authors should discuss what is next for RLT and TAT from the perspective of mCRPC treatment and potential opportunities for cancer therapy. Below are additional comments and suggestions:
Be consistent with the use of acronyms if presented. For example, prostate cancer (PCa) is introduced in the first sentence of the introduction but not use consistently through the manuscript.
Lines 34-37: include the appropriate references for the different type of treatments available for mCRPC.
Line 37-38: “However, survival benefits are limited (3).” This sentence requires additional context regarding how the different treatments are applied. For instance, are the different treatments used as therapy or as palliative care?
Line 38-41: include the respective references showing promising results of RLT and TAT.
Line 47-48: include references for the expression of PSMA in different organs.
Line 53-54: include reference regarding the range of beta particle in tissue
Line 59-61: the sentence regarding the decay of Ac-225 through alpha particle emission should be revised. Also, include the respective references for range of alpha particles in tissue.
Line 64: I am confused by the term “radiolabeled tracers”. What do you mean by it? Both radionuclides are not used at a trace level during therapy or imaging.
Line 72: This is the same sentence used in the abstract
Line 73-74: Why is it important to summarize this data? What kind of perspective are the authors providing? Is this from the medical standpoint?
Line 83: The authors should define I&T for readers without similar background and experience.
Line 101: Please clarify what therapy cessation is in the context of the article and how it was measured.
Line 102-106: What are the contributions of this review article with respect to the systematic review performed by 26.
Line 118: Is 7.4 GBq the total activity received during treatment? Or is the activity per cycle?
Line 118: the sentence “A total of 831 patients was included in this study” lacks context.
Line 142-143: This sentence can be combined with the following one to give better flow to the text.
Line 168-169: Does this sentence correspond to the work done by Von Eyben?
Line 181: there is a typo in “nog”
Line 185: I am confused by this sentence “The pooled HR for patients with PSA decline >50% was 0.53.”. How is this sentence/result different than the previous sentence?
Line 222: What is the perspective of the authors regarding the lack of phase 1 study with Ac-PSMA?
Lines 224-226: This sentence should be revised.
Line 226: This sentence is out of context and does not connect with the previous one. There is a typo on the table, the proper table to reference is Table 3.
Line 282: there is a typo in this sentence “… to patients who with any …”
Lines 294-295: “For these patients, tandem therapy with 177Lu-PSMA combined low activity 225Ac-PSMA might be an alternative treatment option.” Is this sentence the authors’ opinion? Or a reference to Khreish?
Lines 296-300: Are the activities used in this single cycle enough to cause xerostomia (grade 3 or 4) or other side effects? The authors should reflect on some of the values reported in Table 2 and the discussion presented.
Line 342: define mHSPC
Lines 345-346: I reiterate an initial comment, why do the authors think that a phase 1 dose-escalation study has not been performed with Ac-225?
Line 355: “Often xerostomia is the main reason to discontinue treatment”, what are the root causes of xerostomia when using Ac-225?
Line 360: define HRQOL
Reviewer 3 Report
A brief summary:
Recent advances in theragnostic have been showing that alpha and beta emitter radiopharmaceuticals represent an important therapeutic strategy to treat aggressive and invasive tumors. The authors of this review discuss the recent results and the possible expectations on using of Lutetium-177 labeled PSMA (177Lu-PSMA) and Actinium-225 labeled PSMA (225Ac-PSMA) in patients with metastatic castration-resistant prostate cancer.
Broad comments:
The authors describe a series of results of clinical trials regarding the treatment of both radiopharmaceuticals. The review provides a good illustration about the state of the art concerning the use of these two theranostic radiopharmaceuticals. However, the authors should better investigate some aspects and not limit their discussion to a simple list of clinical results. The review lacks a critical point of view about the state of the art and research perspectives by Authors. Moreover, the authors do not compare the advantages and limitations of the two radiopharmaceuticals
Specific comments:
- In the introduction, if any, I would suggest adding more up-to-date epidemiological data than 2018.
- The authors could provide a short summary about the types of prostate cancers. It would be appropriate to describe whether a patients stratification is possible in relation to the grade and to the prognostic outcome beyond CRPC type. The authors could specify the biological function of the PSMA antigen and they could also briefly mention the possible reasons related to the expression increasing in tumor.
- The authors should indicate and discuss the difference between radioligand therapy (RLT) and targeted alpha therapy.
- Lines 59-61: the sentence is not clear.
- Lines 66-68: this sentence would not motivate the use of Lutetium-177. The authors should also mention the advantages, regardless of the LET, in order to persuade the clinical approach of beta emitter radiopharmaceuticals such as 177Lu-PSMA.
- Line 66: I would suggest directly writing “DNA” without the extended form.
- Lines 111-115: I would suggest explaining more clearly this concept.
- Line 115: It is not appropriate to write “Unlike the TheraP trial (will be discussed later)…” in this sentence. At least, the authors should introduce a difference between the two trials before citing them.
- Line 181: typing error “nog”.
- I would suggest specifying the concept of RECIST criteria.
- Authors should rearrange the use of abbreviations and their extended form along the manuscript.